# The High Cost of the Low-Cost Polybag System: A Review of Nursery Seedling Production Systems

Diane L. Haase [1,*], Karma Bouzza [2], Lucy Emerton [3] , James B. Friday [4], Becca Lieberg [5], Arnulfo Aldrete [6] and Anthony S. Davis [7]

1 Reforestation, Nurseries and Genetic Resources, USDA Forest Service, Portland, OR 97204, USA
2 Independent Researcher, Beirut, Lebanon
3 Environmental Economics and Finance, Environment Management Group, Cambridge CB3 9EG, UK
4 College of Tropical Agriculture and Human Resources, University of Hawai'i at Mānoa, Hilo, HI 96720, USA
5 Independent Researcher, Rebecca Lieberg Consulting, Virgin, UT 84779, USA
6 Department of Forestry, Colegio de Postgraduados, Texcoco 56230, Mexico
7 College of Life Sciences and Agriculture, University of New Hampshire, Durham, NH 03824, USA
* Correspondence: diane.haase@usda.gov

**Abstract:** An important strategy for meeting global landscape restoration goals is nursery production of high-quality seedlings. Growing seedlings with attributes that promote post-planting survival and growth can be dramatically influenced by the nursery container system. In many countries, nurseries produce seedlings in polybags filled with excavated soil. These seedlings often develop deformed roots with limited fibrosity which can lead to poor survival and growth after outplanting. Polybags are initially inexpensive but using these single-use plastic containers accrues expenses that are often untracked. Comparisons among nursery production systems must account for factors such as container longevity, labor efficiency, and seedling field performance. A more holistic approach to account for environmental, economic, social, logistic, and cultural elements in the cost–benefit equation that influences nursery production systems is needed. Converting to a modern container system requires concomitant adjustments in nursery scheduling and culturing matched to the new stock type. Doing so provides an opportunity to align nursery production techniques and resulting seedling attributes with anticipated field conditions. This article describes and discusses the advantages and disadvantages of nursery production systems and provides recommendations and case studies to aid nurseries in improving seedling quality toward meeting restoration goals in a cost-effective and timely manner.

**Keywords:** seedling quality; landscape restoration; cost-effective reforestation; target plant concept

## 1. Introduction

Many countries are facing unprecedented environmental issues resulting from years of widespread land degradation. In response, many world leaders have pledged massive landscape restoration goals over the next 10 to 20 years to ameliorate these problems. A primary restoration strategy is to establish plants on degraded lands. While natural regeneration and direct seeding can be effective in some areas, success of these approaches depends on abundant, available seed, favorable environmental conditions for germination and establishment, and low predation from rodents and birds [1,2]. To increase certainty and address these limitations, the strategic use of nursery-grown plants is also necessary to meet restoration goals. If successful in restoring functional, resilient ecosystems, a multitude of environmental, social, and economic co-benefits can result. The restoration or improvement of ecosystem services has a direct, positive impact on human wellbeing [3], and there is now ample evidence of the links between reforestation and livelihoods, including the potential to reduce poverty and to increase socio-ecological resilience and social equity [4]. These benefits, however, are often not fully realized due to suboptimal

reforestation outcomes frequently caused by planting poor-quality seedlings with low potential to thrive, especially on harsh sites.

Too often restoration programs focus on quantity rather than quality of seedlings planted, even though this commonly results in unacceptably low establishment, survival, and growth. In recent years, news stories tout record-breaking numbers of seedlings planted (see, e.g., in [5]), but seedling quality and subsequent field success are often questionable (see, e.g., in [6]). In our collective experience, we have regularly observed many planting projects worldwide using imbalanced, unhealthy seedlings. The quantitative approach to focus on the number of seedlings planted rather than the number of surviving seedlings after a specific evaluation period skews the financial perspective by not accounting for achievement of project objectives. While cost and quality are not synonymous, a comprehensive cost–benefit analysis is critical for total project accounting. Producing higher quality seedlings can require greater resources, but when seedling performance is increased, initial seedling cost can be dramatically offset. For instance, using a simple, hypothetical example, a project with 500 high-quality seedlings, each costing $3 USD to produce and $1 USD to plant and having a field survival of 80% results in a total cost of $2000 USD, or an average of $5 USD per surviving tree. Comparatively, a hypothetical project with 1000 low-quality seedlings, each costing $0.50 USD to produce and $1 USD to plant and having a field survival of only 15% results in a total cost of $1500 USD or an average of $10 USD per surviving tree. Thus, even though the second scenario plants twice as many trees for a lower initial cost, there are fewer surviving trees in the longer term with double the cost for each compared with the first scenario. Note that this simple example does not account for additional costs incurred by delayed restoration benefits, future replanting needs, and growth rate differentials due to initial seedling quality.

Obviously, many variables will influence cost efficiency, but the emphasis on seedling quantity with little regard for seedling quality can ultimately be more costly and compromise the ability to meet restoration goals. For example, Chile established specific forest restoration goals to be met by 2035. A four-year (2016–2019) analysis determined, however, that the goal could not be achieved until 2181 at the current rate and quality of plant production [7]. The shortfall is attributed to lack of expertise, capacity, and resources—a pervasive issue for many nurseries worldwide [8]. To be fully successful, all aspects of the reforestation pipeline—seed, nurseries, outplanting, and post-planting care—must function properly and cooperatively [9]. If one part of the reforestation pipeline fails, investments into other parts will also be lost. While this paper focuses on the technical aspects of reforestation success, local social, political, and economic factors also play an important, and often dominant, role in determining success. Because landscape-scale reforestation will, in most cases, impact local communities, it is critical that local communities perceive the project as providing opportunities for locally valuable goods and services rather than as a threat [10,11].

Seedling quality is primarily determined by the nursery growing system and culturing techniques. Ideally, these factors work together to meet specific phenological, genetic, physiological, and morphological targets matched to species' growth patterns, outplanting site conditions, and land manager's objectives such that the right tree is planted in the right place [12]. For nurseries producing container-grown seedlings, the type of container and the growing medium they use profoundly affect seedling development and quality. Nursery containers vary by volume, dimensions, shape, drainage hole positions, and presence (or absence) of slits or ribs. These characteristics influence growing density, irrigation needs, and plant size and development [13]. A quality growing medium provides support to the plant, has suitable physical characteristics (aeration, porosity, water-holding capacity, and bulk density) and chemical characteristics (fertility, pH, and cation-exchange capacity), and is free of pests and pathogens [14,15]. In locations where resources are limited, nurseries typically use polybag containers and locally available soil due to low initial cost. In other locations, nurseries use modern containers and soilless media (substrates). The objective of

this article is to describe and compare the polybag and modern container growing systems in terms of seedling quality, nursery efficiency, overall economics, and outplanting success.

## 2. The Polybag System

Polybags, made from polyethylene plastic (usually black), are used worldwide because they are inexpensive (approximately $0.01 to $0.05 USD each, depending on size and vendor), lightweight, collapsible, cost less to ship compared with modern nursery containers, and are familiar to nursery managers. Polybags are available in multiple sizes. As with other containers, larger bags generally produce larger seedlings [16] given time, nutrients, and water. Additionally, copper-coated polybags are available in some locations and can reduce root deformations by chemically pruning roots [17,18].

Polybags have smooth surfaces. As a result, when seedling roots reach the polybag wall, they spiral around and tend to concentrate in the bottom of the bag (Figure 1 [19,20]). This deformed root growth pattern results in low overall fibrosity and volume [18]. Root deformation from spiraling is exacerbated when seedlings are not outplanted in a timely manner. Deformed roots can lead to reduced growth, poor resistance to stress, and early dieback [21]. Seedlings with spiraled roots may not adequately anchor the plant and may have reduced water and nutrient uptake after planting [22]. The snarled root mass can even cause girdling [23] and instability that may result in windthrow years after outplanting.

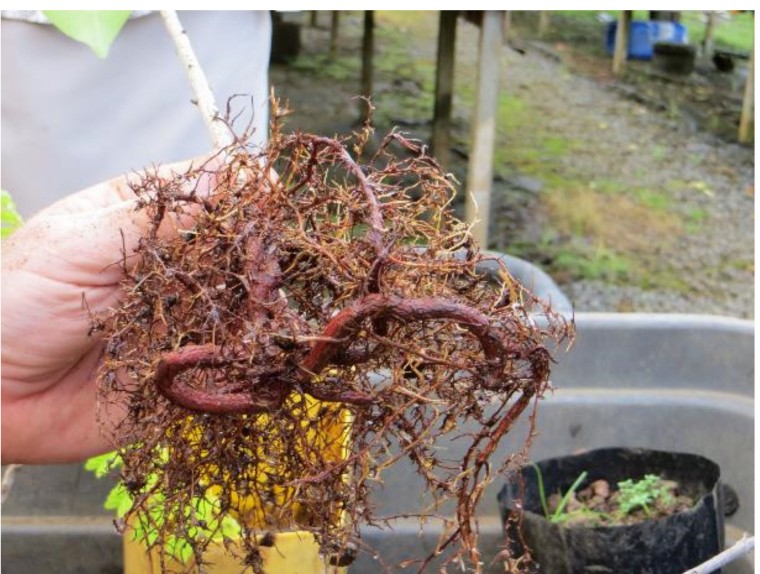

**Figure 1.** Seedlings grown in polybags often have spiraled roots that are concentrated in the bottom of the bag. Photo by Diane L. Haase.

During nursery cultivation, polybags are often placed on the ground where roots can grow into the soil through the drainage holes or tears in the plastic resulting in root damage or loss when seedlings are picked up to be sent to the field (Figure 2). This placement can also be ergonomically challenging for nursery workers. The combined effect of root deformation, low fibrosity, and damage reduces the functional root area and decreases the root-shoot ratio. Such an imbalance reduces plant performance due to a mismatch between the shoot's transpiration and nutrient demands and the root's uptake capabilities, especially on drier sites [24].

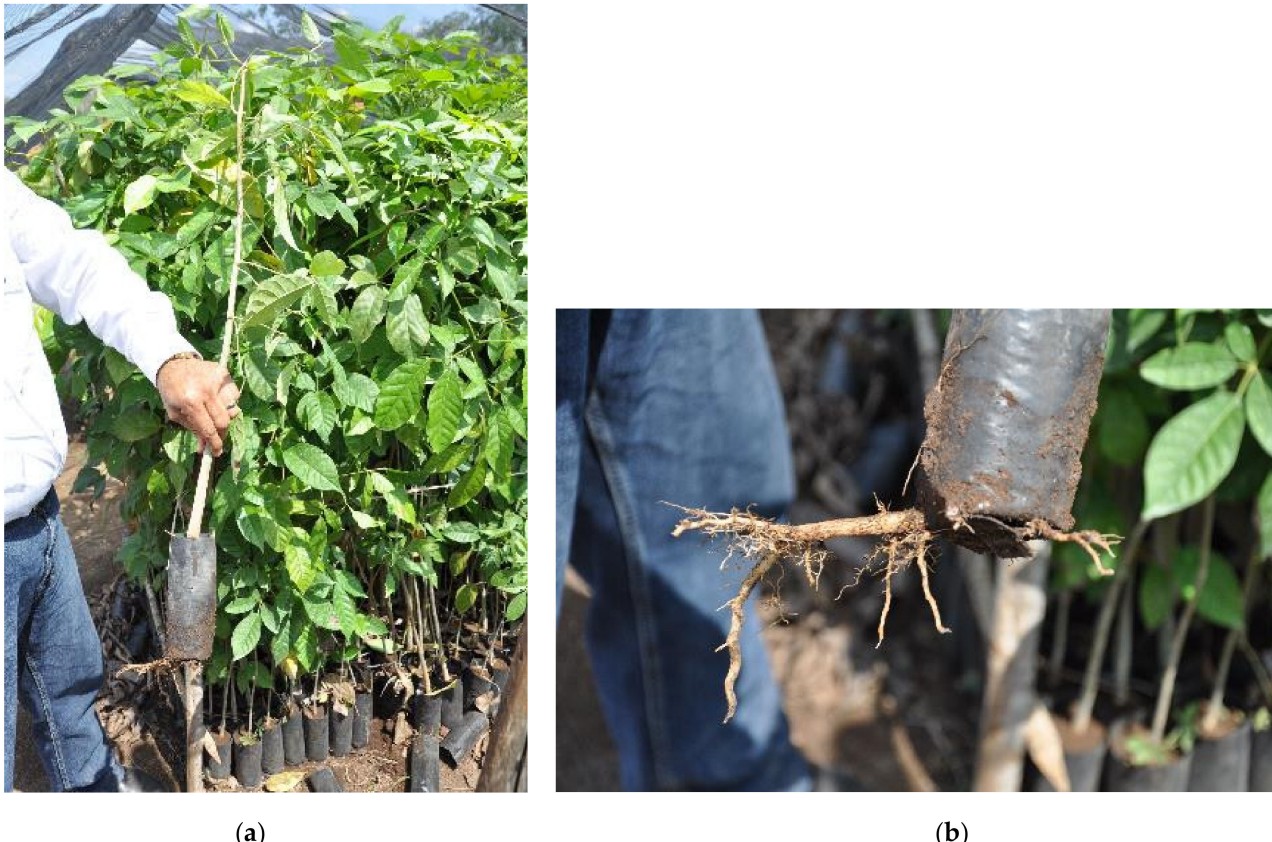

(**a**)  (**b**)

**Figure 2.** Nurseries using polybags often fill the bags with field soil and grow the seedlings directly on the ground (**a**). Often, seedling roots push through the bag and grow into the ground, especially if they are grown in the container for too long. Root loss and damage can occur when seedlings are removed from the nursery (**b**). Photos by Arnulfo Aldrete.

Polybags are usually filled with field soil collected within a short distance from the nursery. Field soil is often free except for the labor and transportation. Typically, soils excavated for nursery production have high clay content, high bulk density, low fertility, and poor drainage. As a result, soil can be quite heavy and bulky, thereby requiring more labor and space for handling, transporting, and planting compared with soilless growing media. Field soil can also carry weed seeds, insects, and diseases. Excavated soil loses its structure and is prone to compaction and waterlogging (Figure 3), which impairs root development and architecture. Some nurseries mix sand, manure, compost, or other materials with the soil. Sand is often added to the soil with the intention to improve drainage. The opposite effect can occur, however, depending on the ratio of sand to clay and the presence or absence of organic material. Excavating field soil for nursery production is an unsustainable practice and can cause erosion and site degradation. When soil is excavated for nursery use, the topsoil is quickly depleted and can take many decades to recover, thus creating more areas in need of restoration [21].

Although initial cost for polybags is relatively low, polybags are typically a single-use container and must be purchased annually. Worldwide, plastic pollution is recognized for its harmful environmental, social, and economic impacts [25]. Many nations have established policies and enacted legislation to reduce single-use plastics [26,27]. In fact, polybags for nursery production have already been banned in a few countries [28,29]. Furthermore, contributing to the waste stream (along with mining soil) can be viewed in conflict with nurseries' principles toward environmental restoration.

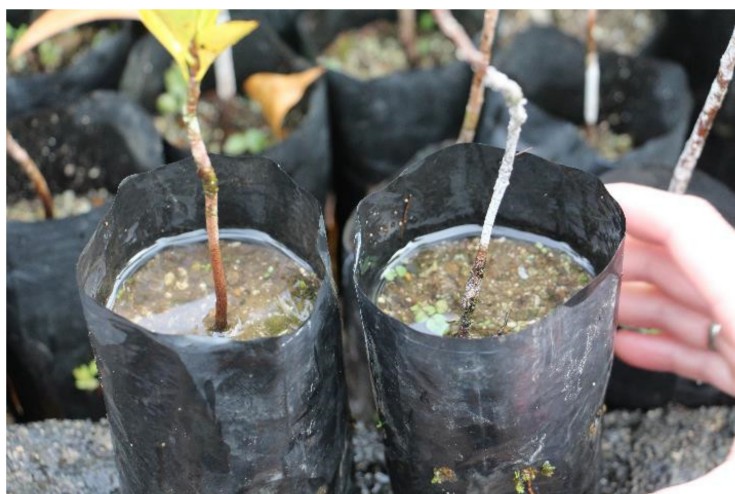

**Figure 3.** When using field soil in polybag containers, water can drain poorly and result in waterlogging. Photo by J.B. Friday.

## 3. The Modern Container System

Modern containers are available in individual cells (Figure 4a), free-standing containers (Figure 4b), or aggregate blocks (Figure 4c) and range in size, shape, material, and cost [13] (Table 1), allowing for customization to meet a wide diversity of production variables. For example, fixed cells minimize the need for handling individual units, whereas containers consisting of individual cells can be moved within a tray or rack allowing for easy sorting and spacing in response to variable germination or growth (Figure 5). Most modern containers have rigid walls with internal vertical ribs or slits to direct root growth downward, thereby preventing spiraling and promoting good root fibrosity and architecture (Figure 6). In a study to simulate modern containers, plastic beverage bottles were modified by adding slits or ribs and were successful in reducing root spiraling compared with polybags [30]. Modern containers also have bottom holes that ensure adequate drainage and provide for natural air pruning to inhibit root growth out of the containers. Although initial cost and shipping per modern container is higher than polybags, they can be reused for 5 to 12 years depending on the container material and the number of nursery production cycles (Table 1).

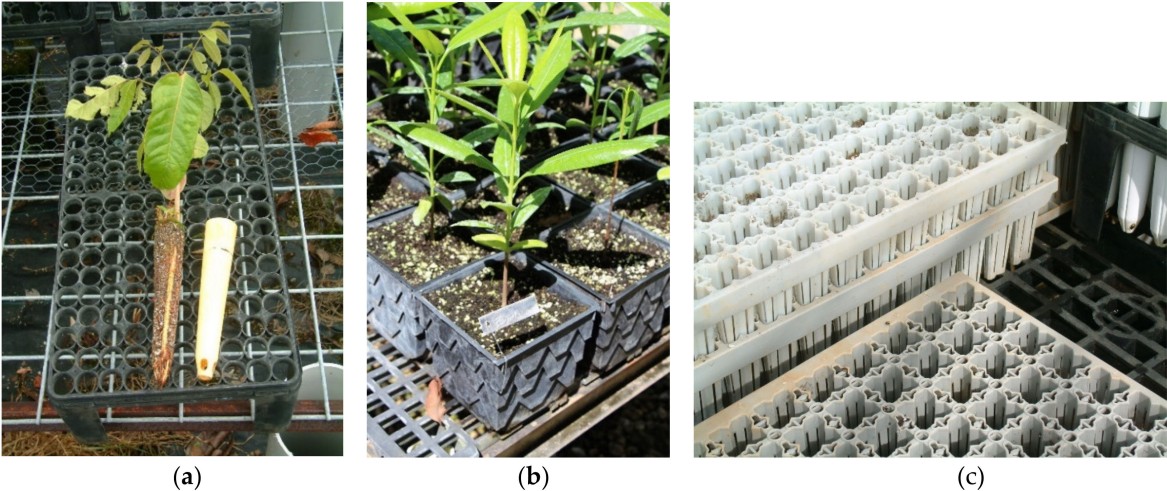

(**a**)　　　　　　　　　　　　(**b**)　　　　　　　　　　　　(**c**)

**Figure 4.** Modern container types vary widely by dimensions and design. Common types are (**a**) individual cells, (**b**) free-standing containers, and (**c**) molded trays with multiple cells. Photos by J.B. Friday.

**Table 1.** Characteristics and costs of some common modern container types and sizes. Prices (USD) courtesy of Stuewe and Sons, Inc., USA 2021 based on a single case; large quantity orders result in approximately 10% price reduction (additional container details available at https://stuewe.com [accessed on 5 August 2021]).

| Container Type [1] | Cell Volume (cm³) | Density (#/m²) | Expected Longevity (Years) | Cells per Tray/Block | Containers per Case | Container Cost per Case (USD) | Trays per Case | Tray Cost per Case (USD) |
|---|---|---|---|---|---|---|---|---|
| *Book containers* | | | | | | | | |
| Rootrainer™ Tinus Z10 [2] | 350 | 298 | 2–4 | 4 cells per book/ 9 books per tray | 200 books | 475 | 50 | 200 |
| *Containers with exchangeable cells held in a tray or rack* | | | | | | | | |
| Deepot™ D40H + D20 tray | 656 | 174 | 10–12 | 20 | 315 cells | 107 | 10 | 80 |
| Deepot™ D60H + D20 tray | 983 | 174 | 10–12 | 20 | 210 cells | 86 | 10 | 80 |
| Ray Leach Cone-tainer™ SC10U + RL98 tray | 164 | 527 | 10–12 | 98 | 1100 cells | 220 | 10 | 93 |
| Ray Leach Cone-tainer™ SC7U + RL98 tray | 107 | 527 | 10–12 | 98 | 1650 cells | 248 | 10 | 93 |
| Treepot™ TP414 + Tray 10 | 2830 | 82 | 4–6 | 9 | 210 pots | 105 | 20 | 88 |
| *Blocks (trays) composed of cavities or cells* | | | | | | | | |
| Rootmaker® RM18T | 410 | 118 | 5–6 | 18 | N/A | N/A | 25 | 115 |
| Hiko™ V265 | 265 | 368 | 12–15 | 28 | N/A | N/A | 10 | 60 |
| Hiko™ V530 | 530 | 198 | 12–15 | 15 | N/A | N/A | 10 | 59 |
| Styroblock™ V77170EXT [3] | 164 | 365 | 7–10 | 77 | N/A | N/A | 6 | 51 |
| T.O. Plastics TO50SR5 tray | 195 | 333 | 3–5 | 50 | N/A | N/A | 25 | 129 |

[1] Manufacturers: Rootrainer, Deepot, Ray Leach, and Treepot (Stuewe & Sons, Inc., Tangent, OR, USA); Rootmaker (Rootmaker Products Company, LLC., Huntsville, AL, USA); Hiko (BCC AB, Landskrona, Sweden); Styroblock (Beaver Plastics, Ltd., Alberta, Canada); T.O. Plastics (T.O. Plastics, Clearwater, MN, USA). [2] Rootrainers ship flat and take up relatively little space. [3] Styroblocks are bulky to ship internationally.

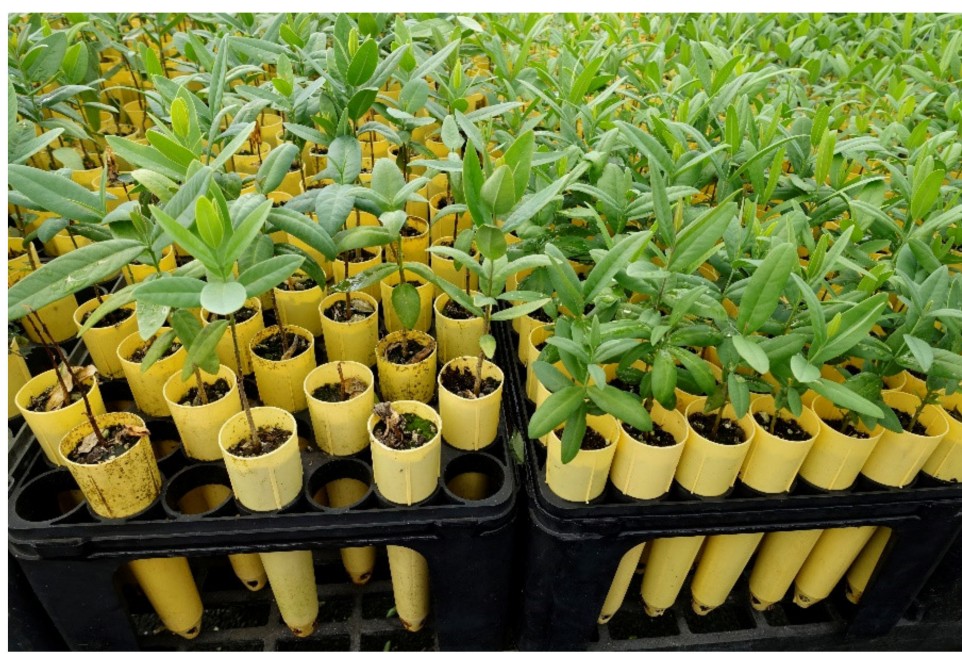

**Figure 5.** Modern container systems consisting of a tray with removable cells can facilitate sorting and spacing as needed in the nursery. Photo by Diane L. Haase.

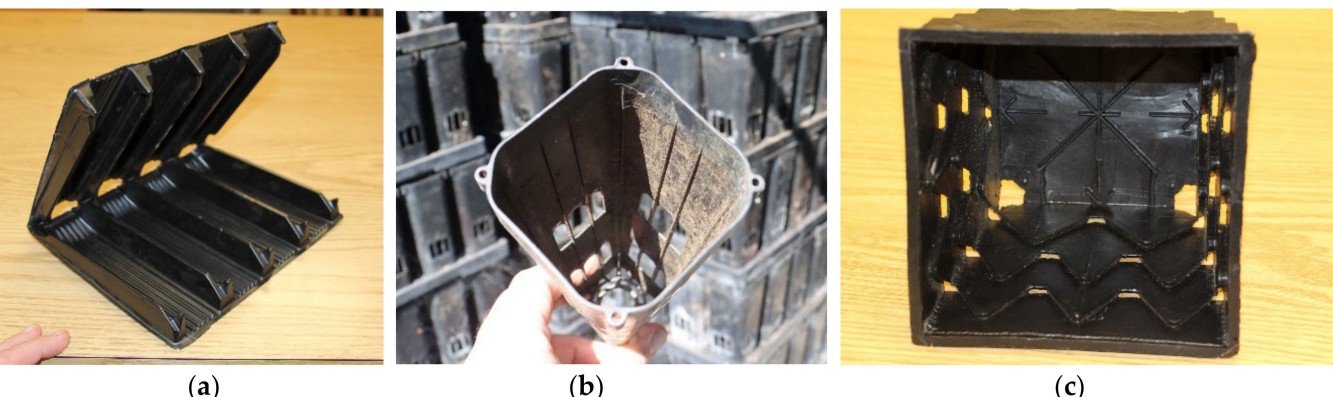

**Figure 6.** Most modern containers are designed with features to encourage quality root architecture such as (**a**) internal grooves, (**b**) internal ribs and holes, and (**c**) ridges. Photos by J.B. Friday.

Seedlings grown in modern containers have comparatively uniform growth with more vigorous roots and better root–shoot balance, and they are often ready for planting when they are much smaller than those grown in polybags [31]. The smaller size and shorter growing time reduce the amount of space, labor, growing medium, and other resources needed in the nursery [21]. Smaller seedlings are also relatively lightweight and thus easier to carry, transport, and outplant. After planting, seedlings grown in modern containers tend to have higher long-term growth and survival rates than those grown in polybags [19].

Modern container systems typically use soilless substrates, partly because these smaller volume containers can be difficult to fill with field soil. Substrates usually consist of mixtures of organic and inorganic materials. These substrates have superior physical and chemical characteristics for nursery production compared with soil [14] including adequate porosity and aeration, high cation exchange capacity, and low bulk density. These media are also initially free of pathogens and weed seeds. Although soilless growing mixes can be more expensive than soil, modern containers usually have lower volume and therefore require less substrate per seedling.

In North America and Europe, *Sphagnum* peat moss is commonly the main organic component used in container plant production due to its favorable characteristics. Peat extraction, however, rouses environmental concerns regarding sustainability and impact. Furthermore, peat is not readily available in some parts of the world and shipping is cost-prohibitive. Several alternative organic materials can be used to create renewable, locally available growing media for plant cultivation [32,33]. These alternatives include compost [34,35], coconut husk [36–38], bark [39], rice hulls [40,41], sawdust [42,43], biochar [44], sludge [45], solid anaerobic digestate [46], and other materials. These organic materials can be mixed with inorganic components such as vermiculite and perlite.

Even though modern containers cost more initially, this can be more than offset by their higher longevity, better labor efficiency, increased seedling quality, and improved outplanting performance. In a study to compare container cost efficiencies [47], labor to fill polybags was 5.3 times higher compared with filling the same number of seedlings grown in Hiko™ trays (durable, reusable plastic trays with multiple built-in cells). Each polybag required 425 cm$^3$ of potting mix compared with only 100 cm$^3$ required for each Hiko™ cell. Furthermore, labor for potting, watering, hauling to the field, and outplanting was 2.0, 2.5, 3.2, and 1.9 times greater, respectively, for polybag seedlings compared with Hiko™ seedlings. With these efficiencies, the overall labor cost per hectare was reduced by 8.42 person-days, a savings many times greater than the initial container cost.

## 4. Comparison Challenges

When upgrading to a modern growing system, concomitant adjustments to the nursery's culturing regime, including watering, fertilization, and spacing, must occur to fully realize the benefits of the new system (see "Recommendations" section). Otherwise, nursery personnel may conclude that polybags and/or field soil are superior based on confounding factors [48] and may produce low-quality seedlings in the new system. For example, one study found that shoot and root growth of *Pinus occidentalis* was greater for seedlings grown in polybags than for those grown in D40 Deepot™ containers, but the comparison was confounded by the polybags having 50% greater volume [49]. Even still, the authors noted that the higher root–shoot ratio of D40 seedlings compared with polybag seedlings was potentially more favorable for dry outplanting conditions. In another study, researchers found higher physiological function of *Hevea* seedlings grown in polybags compared with Rootrainers™ and attributed this to "root confinement" [50]. This finding indicates that seedlings were grown too long in the Rootrainers™ and the results were due to a mismatch between the growing schedule and the container size. In yet another example, researchers concluded that seedlings grown in polybags had better performance based on germination and shoot characteristics than those grown in Rootrainers™ [51]. Germination, however, is primarily a function of moisture and temperature immediately surrounding the seed and should not be influenced by container type. Thus, these results suggest that irrigation frequency may have been insufficient for the Rootrainers™ which dry more quickly than polybags due to the smaller volume of growing medium. Despite the authors' conclusions, however, their data showed significantly greater root development relative to container size for seedlings grown in Rootrainers™ compared with those grown in polybags. When grown properly, seedlings grown in Rootrainers™ have greater survival, growth, and root–shoot ratio compared with those grown in polybags [31].

## 5. Economic and Environmental Consequences and Opportunities Associated with Nursery Growing Systems

Unsuccessful restoration is costly. Wastage of seedlings, labor, and other consumables is just one element of this cost. The direct costs of afforestation and reforestation are almost always dwarfed by the immense social and economic value of the ecosystem services that are generated by healthy forest landscapes (and, conversely, by the foregone benefits or resulting costs, losses and damages that occur when restoration efforts underperform or fail). Such values are now well documented. The estimated annual cost of land degradation due to land use and land cover change is more than $231 billion USD, or about 0.41% of global GDP [52]. Direct physical products and raw materials account for less than half of this figure; the majority comes from the loss of non-market, ecosystem services such as watershed protection, carbon sequestration, genetic diversity, and cultural services. Such costs and damages directly impact forest and agricultural land and resource users' livelihoods and have wide-ranging effects on the many other groups that depend on ecosystem services such as water quality and flow regulation, wildlife habitat, crop pollination, pest control, seed dispersal, disaster risk reduction, climate adaptation and mitigation, and so on. Livelihoods, health, and nutrition for more than 20% of the world's population depend on forests [53], including more than 90% of people living in extreme poverty [54]. Furthermore, forested watersheds and wetlands supply as much as 75 percent of the world's accessible fresh water for domestic, agricultural, industrial, and ecological needs, including 90% of the world's largest cities [55].

When the wider ecosystem service benefits are considered, the return on investments in forest restoration are extremely high. For example, in Portland, OR, Portland, ME, and Seattle, WA, every $1 USD investment into forest watershed protection can save $7.50 to nearly $200 USD in costs for new water treatment and filtration facilities [56]. In the Global South, these figures tend to be even higher. For example, every $1 USD invested in landscape conservation and restoration in the Upper Tuul ecosystem in Mongolia can generate $15 USD for downstream domestic and industrial water users in Ulaanbaatar [57],

while public investments in Myanmar's natural forests show a 1:40 economic multiplier in terms of ecosystem service benefits generated for the wider economy and population [58].

Just as the economic returns to tree planting and forest landscape restoration extend well beyond direct income and earning opportunities, the cost–benefit equation that determines container choice in nurseries is also far more complex than simply comparing market prices and survival rates. Other important issues and considerations include availability, transport and distribution costs, and even taxes and import duties [13]. These issues are often of particular concern in the Global South, where forest managers and farmers are only able to access very limited market choices. Polybags are often the most easily obtainable container for many rural nurseries. In many cases, there has not yet been sufficient investment in producing and marketing other products—even when it would be more cost-effective, the raw materials are readily available locally, and doing so could provide significant opportunities for local employment and income generation.

To overcome these issues, nurseries must be recognized for their fundamental role in achieving restoration goals and must have consistent and sustained technical and financial support to enable the development and promotion of economically and environmentally beneficial technologies [8,59,60]. This support, however, is often politically motivated and rarely factors in the broader economic impacts associated with avoided environmental damages and potential for additional, often local, value-added benefits. Decision-makers tend to emphasize direct costs and market expenditures rather than benefits or avoided losses and damages. Environmental externalities and full lifecycle impacts are rarely factored into the price of traditional plastics, which remain artificially low, meaning that the relatively low market price of a polybag does not fully reflect its true cost to society and the broader economy [61]. Such a situation perpetuates the use of polybags by encouraging the greatest short-term quantity for the lowest initial price regardless of the long-term outcome or the broader economic impacts and consequences.

A wide range of market support mechanisms, incentives, and investments—as well as a greater political and policy priority—could potentially redress these imbalances and serve to encourage and enable nurseries and forest managers to shift to alternative containers. For example, policy incentives such as payments for environmental services have been proposed as an instrument to encourage the adoption of biodegradable bags among small-scale nursery operators in Africa [62]. Any changes in production technologies and inputs must, however, make financial and business sense to the landholders and enterprises that are involved in producing and raising seedlings. Improved project evaluation could also encourage project managers to invest into quality nursery production. If success is measured by number of seedlings produced regardless of quality and subsequent growth and survival, then managers will continue to produce seedlings as cheaply as possible. If, however, success is measured by survival and growth of outplanted seedlings with recognition of the long-term benefits associated with wider ecosystem services, then project managers would have more incentive to invest in high-quality stock.

## 6. Case Study—Lebanon Reforestation Initiative

In the last several decades, Lebanon has experienced an increased rate of deforestation due to exploitation of forests for wood, agriculture practices, and quarrying activities leading to a drastic decrease of green cover [63]. Urbanization and urban sprawl caused by a history of civil unrest and increased costs to live in cities have also contributed to deforestation [64]. In addition to climate change mitigation, Lebanon's forest ecosystems provide local communities with a wide range of services including fauna and flora conservation, fresh water; fuel wood; edible and medicinal plants; as well as spiritual, recreational, aesthetic, and educational opportunities [65]. Thus, sustaining reforestation and restoration efforts is a national priority.

The Lebanon Reforestation Initiative (LRI) was initiated by the U.S. Department of Agriculture Forest Service in 2011 with funding from the U.S. Agency for International Development and was registered as an NGO in 2014. LRI's primary objective is to improve reforestation practices in support of national efforts to sustainably increase green cover in Lebanon. A focus for LRI was to produce robust tree seedlings with increased survivability after planting [66]. To that end, several factors were considered including container selection, growing media, propagation environment, timing, seed treatments, irrigation and fertilization management, monitoring, and other practices.

The Deepot™ D40 container (656 mL; Stuewe and Sons, Inc., Tangent, OR, USA) was selected for the project due to its similarity to the depth of commonly used polybags and the depth of soil at potential outplanting sites. Pots were filled with a 50:50 mixture of peat:cocopeat with some amendments (Table 2). With the support of an economist, data were collected and evaluated from nine native tree nurseries across Lebanon to compare container and growing media costs between the "old" and "new" nursery systems (Table 2). By accounting for the fact that modern D40 containers can be used for 10 years, the annual material costs were similar for each stock type. Previously, seedlings were produced in polybags filled with soil and top dressed with perlite. Field soil, while readily available locally, required collection, transportation, and sterilization treatment before use. Excavated field soil had poor aeration and was a source of disease. This "old" system had low germination and failed to reach target quality before planting (Figure 7). After planting, field survival averaged approximately 20%.

**Table 2.** Cost comparison by the Lebanon Reforestation Initiative for containers and growing media for one growing season. The "old" system consisted of polybags and field soil and the "new" system used modern containers (Deepot™ D40) and a soilless substrate. The cost of D40 containers and trays were divided by 10 years based on expected longevity when well maintained. Other costs (such as seed collection and treatment, labor, infrastructure, irrigation, land rent, and maintenance) were considered non-variables between polybag and modern container growing systems. Prices are from 2012.

| | "Old" Nursery System | | "New" Nursery System | |
|---|---|---|---|---|
| | Cost/Bag (USD) | Cost per 50,000 (USD) | Cost/Cell (USD) | Cost per 50,000 (USD) |
| Containers | | | | |
| Polybags (single use) | 0.0555 | 2775 | | |
| D40 (used over 10 years) | | | 0.0249 | 1245 |
| D20 Trays (one per 20 seedlings; used over 10 years) | | | 0.0236 | 1180 |
| Growing media | | | | |
| Field soil | 0.1250 | 6250 | | |
| Perlite | 0.0190 | 950 | 0.0125 | 625 |
| Cocopeat | | | 0.0123 | 615 |
| Peat moss | | | 0.0354 | 1770 |
| Basacote plus | | | 0.0200 | 1000 |
| Stockosorb | | | 0.0656 | 3280 |
| **TOTAL** | **0.1995** | **9975** | **0.1943** | **9715** |

LRI started providing technical support to native nurseries in Spring 2012 to assist in shifting their production from the polybag system to science-based practices using Deepot™ containers and the Target Plant Concept [12,67]. Seedling quality in the nursery increased dramatically (Figure 8). In addition to improved nursery practices, LRI developed a field monitoring protocol to assess post-planting seedling survival and identify causes of mortality. For the first reforestation phase, LRI monitored seedlings grown in polybags (2011) and in Deepot™ containers (2012 and 2013) at four sites (Table 3). The improved seedling quality resulting from a shift in nursery practices was evident in the increased field survival at all four sites (Table 4).

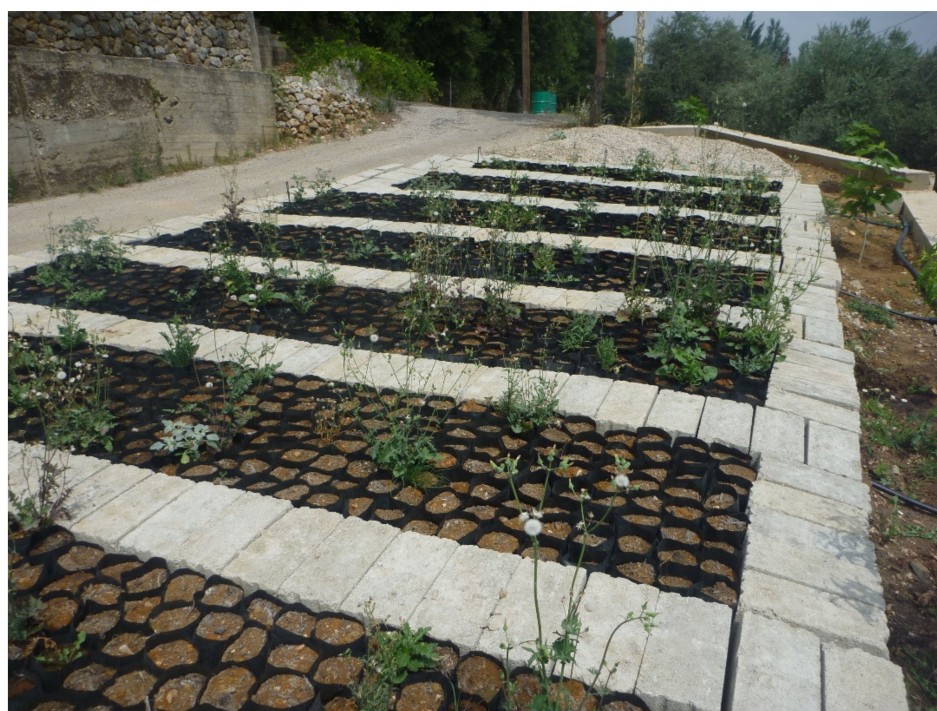

**Figure 7.** Before the Lebanon Reforestation Initiative was established, seedlings grown in polybags with field soil had poor germination and growth in the nursery. Photo by Anthony S. Davis.

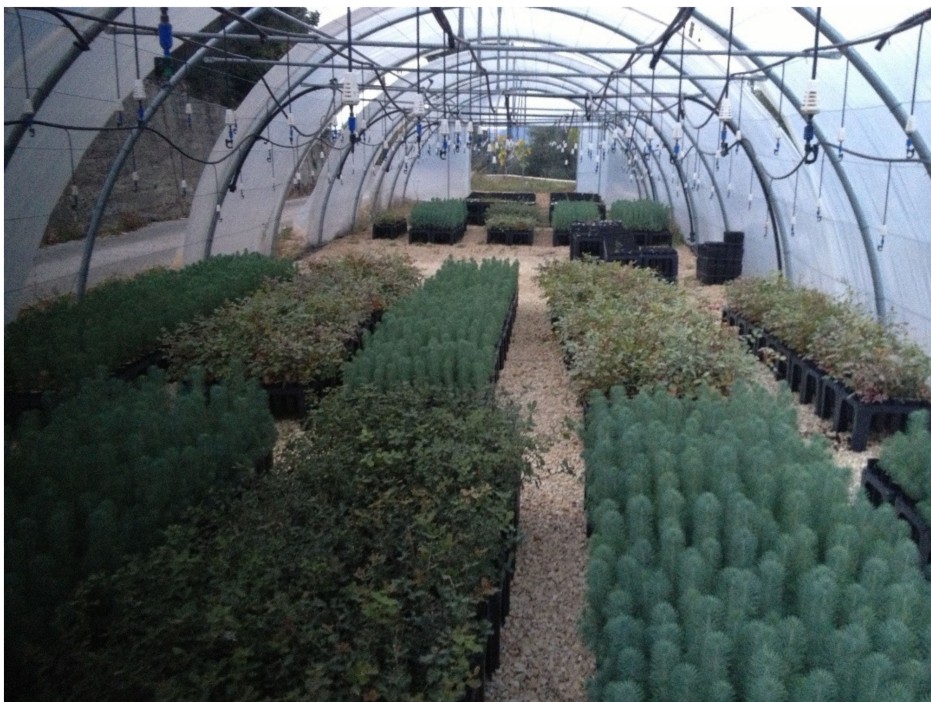

**Figure 8.** The Lebanon Reforestation Initiative program established improved nursery practices which resulted in production of high-quality seedlings that survived and grew well after outplanting. Photo by Anthony S. Davis.

**Table 3.** Descriptions for each of the four sites where seedlings were planted during Phase 1 of the Lebanon Reforestation Initiative program.

| Site | Anjar | Qlayaa | Rashaya | Tannourine |
|---|---|---|---|---|
| Coordinates | 33°44′00.64″ N 35°56′50.33″ E | 33°20′33.96″ N 35°34′11.06″ E | 33°29′06.54″ N 35°51′24.74″ E | 34°12′28.06″ N 35°56′20.67″ E |
| Elevation | 800–900 m | 550–650 m | 1200–1500 m | 1750–1800 m |
| Mean annual precipitation | 200 mm | 600 mm | 400 mm | 1200 mm |
| Rockiness | High | High | Medium to high | Medium to high |
| Soil type | Lithic leptosol | Eutric cambisol | Lithic leptosol | Vertic luvisol |
| Species planted | *Acer* sp. *Amygdalus* sp. *Arbutus andrachne* *Cedrus libani* *Celtis australis* *Cercis siliquastrum* *Cupressus sempervirens* *Fraxinus angustifolia* *Laurus nobilis* *Pinus brutia* *Pinus pinea* *Prunus ursina* *Pyrus syriaca* *Quercus calliprinos* *Quercus infectoria* | *Acer* sp. *Amygdalus* sp. *Arbutus andrachne* *Celtis australis* *Cercis siliquastrum* *Fraxinus angustifolia* *Laurus nobilis* *Pinus brutia* *Pinus halepensis* *Pinus pinea* *Pistacia palaestina* *Prunus ursina* *Quercus calliprinos* *Quercus infectoria* | *Acer* sp. *Amygdalus* sp. *Arbutus andrachne* *Cedrus libani* *Celtis australis* *Cercis siliquastrum* *Fraxinus ornus* *Laurus nobilis* *Pinus brutia* *Pinus halepensis* *Pinus pinea* *Pistacia palaestina* *Prunus ursina* *Pyrus syriaca* *Quercus calliprinos* *Quercus cerris* *Quercus infectori* | *Acer* sp. *Amygdalus* sp. *Cedrus libani* *Celtis australis* *Juniperus excelsa* *Sorbus flabellifolia* *Sorbus torminalis* *Pyrus syriaca* *Quercus brantii* *Quercus calliprinos* *Quercus cerris* |

**Table 4.** Seedling survival at four outplanting sites during the first phase of the Lebanon Reforestation Initiative increased significantly when nursery practices shifted from a polybag system with field soil to a modern container system with a soilless substrate.

| | Survival (%) | | |
|---|---|---|---|
| Site | 2011–2012 (Polybag) | 2012–2013 (D40) | 2013–2014 (D40) |
| Anjar | 18 | 59 | 91 |
| Qlayaa | 42 | 96 | 84 |
| Rashaya | 58 | 69 | 73 |
| Tannourine | 30 | 48 | 61 |

Seasonal nursery and field losses (based on 50,000 seedlings) were calculated to compare old and new nursery practices using the following simple equations:

- Nursery loss = 50,000 seedlings * % mortality * cost of production per seedling
- Field loss = 50,000 seedlings * % mortality * cost of planting per seedling

These calculations showed significantly reduced seasonal losses to nursery and field investments due to reduced mortality of seedlings grown in the D40 containers with improved nursery practices compared with those grown in polybags (Figure 9).

Note that seedling survival rates do not depend solely on good nursery and outplanting practices. Seedling survival on LRI sites have also been influenced by fires, anthropogenic pressures, agricultural activities, and other biotic and abiotic factors. Nonetheless, this study shows that improving nursery practices reduces the overall cost of reforestation. Reducing the need to re-plant large areas due to high annual rates of post-planting seedling mortality leads to reduced overall reforestation costs in Lebanon.

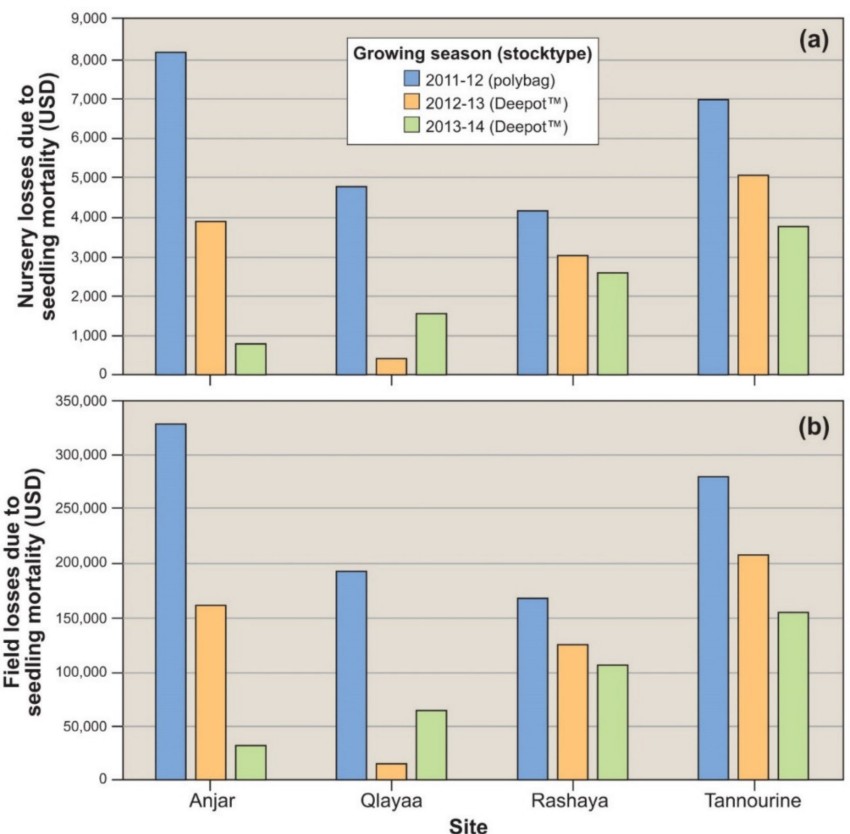

**Figure 9.** Seasonal losses due to mortality after outplanting based on (**a**) seedling production costs and (**b**) planting costs were much higher for seedlings grown in polybags with field soil compared with those grown in Deepot™ containers with a soilless substrate.

## 7. Recommendations for Starting or Converting to a Modern Nursery System

When establishing a new nursery or converting an existing nursery from the polybag system to a modern system, several aspects must be considered to succeed in producing high-quality plants with the highest potential to survive and thrive after outplanting. Although existing nurseries may have structures, water supplies, and personnel already in place, converting from a polybag system should be viewed as a completely new beginning. Several freely available nursery manuals can provide guidance for establishing and refining best nursery practices (see, e.g., in [68–70] and https://rngr.net, accessed on 5 August 2021).

Conversion to a modern container system requires sustained technical support, education, assistance, and encouragement. In Mexico, it was found that 70% of seedlings were still being grown in polybags despite efforts a decade earlier to modernize nurseries [71]. Many of these nurseries had significant problems with infrastructure, labor, growing media, and training. After another decade, approximately 75% or more of seedlings were being grown in modern containerized systems but seedling quality and crop uniformity were often unsatisfactory due to inadequate knowledge of seedling biology, species diversity, and nursery cultural practices necessary to grow high-quality seedlings [71]. Similarly, a national policy to regulate seedling quality in the Philippines was ineffective due to many factors including limited resources and lack of knowledge [72]. Thus, the ability to grow quality seedlings depends not just on having the proper equipment and supplies but also on having ample guidance, knowledge, and motivation.

### 7.1. Container and Substrate Selection

Container selection for a particular plant species is determined by root system morphology, growth rate, length of growing season, seed size, target plant criteria, and outplanting site conditions. For example, plant species that develop long taproots grow best in

tall containers, whereas those with shallow, fibrous roots grow better in shorter containers and those with thick, fleshy roots grow better in wide containers [13]. Similarly, shorter containers are better for plants to be outplanted on sites with shallow soils. Containers consisting of individual cells in racks can facilitate sorting for uniformity, removal of empty cells, and spacing as plants increase in size. Because empty cells can be removed, these containers are ideal for direct sowing which can reduce labor costs and root deformation associated with transplanting from germination trays.

Modern containers can be reused for up to 12 years (Table 1). In between uses, the containers need to be thoroughly washed to avoid passing pests and diseases from one crop to the next. If the nursery gives the container to the customer with the plant, they must establish a system to get the containers back, such as charging a refundable deposit. Alternatively, they can remove the plants from the containers and protect the root plugs from desiccation by bundling and wrapping in burlap, plastic, or other material. In either case, customers must be instructed to plant the seedlings as soon as possible and to provide care for them during the vulnerable stage between nursery and outplanting.

Nursery growing media can be formulated using a variety of organic and inorganic materials [14]. Because of its bulk, all or most of the growing media components would be ideally sourced locally. In many locations, suitable commercial mixes may be available. Compost is a particularly attractive bioresource because a variety of raw materials tend to be readily and locally available and because composting repurposes waste materials that may otherwise be problematic for disposal [33]. Variations in the raw materials and the composting process, however, create variations in physical, chemical, and biological properties of the compost. Thus, some nurseries set up their own composting system to control these variations and to ensure consistent availability.

When developing new growing media, it is recommended to evaluate drainage, fertility, compaction, and plant development throughout the growing season and adjust the growing media formulation as needed for subsequent crops. Nurseries often have more than one substrate formulation. For example, newly rooted cuttings are best struck in a very porous medium to allow good aeration for root formation whereas media for seed propagation should have a finer texture to maintain adequate moisture for germination.

*7.2. Scheduling and Culturing*

Because of the lower container volume and improved substrate characteristics, seedlings grown in modern container systems require more frequent irrigation than those grown in polybags filled with soil. Daily monitoring is critical to ensure growing media do not dry out. One useful technique is to develop irrigation schedules based on container weights. This method results in adequate water availability for a specific container volume and growing medium combination [73]. Likewise, fertility must be adjusted according to the substrate's properties.

Seedlings grown in modern containers with lower volumes are ready for outplanting earlier and have less flexibility for outplanting delays compared with seedlings grown in polybags. This shorter timeframe means greater savings in labor and other resources but requires clear communication between the nursery manager and the land manager to establish well-defined timelines for production and planting. Seedling readiness to leave the nursery must be matched with demand for outplanting to prevent seedlings from becoming overgrown.

Seedlings in modern containers are best kept on elevated platforms or benches with mesh or slat surfaces to improve air pruning of roots (Figure 10). Setting the nursery up in this manner also facilitates good sanitation practices, protects containers from breakage, and reduces ergonomic risk to nursery staff compared with containers placed on the ground.

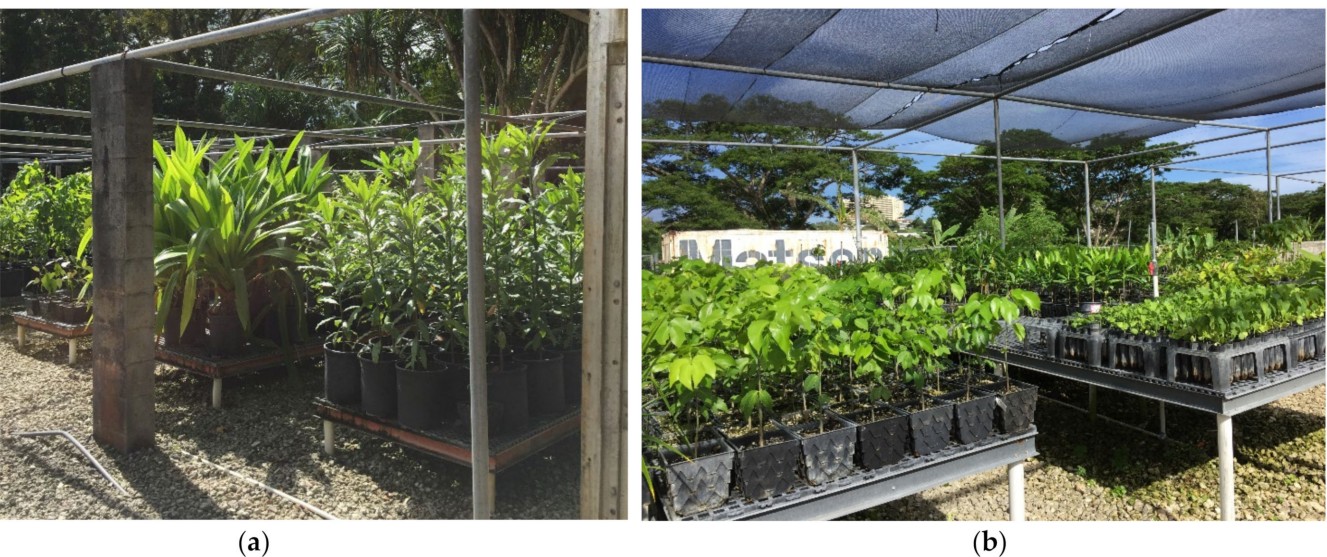

(**a**)　　　　　　　　　　　　　　　　　　　　　(**b**)

**Figure 10.** Placing seedling containers on (**a**) elevated platforms or (**b**) benches promotes air pruning and drainage, improves ergonomics for nursery staff, and can be part of a nursery sanitation program to control pests and diseases. Photos by Diane L. Haase.

### 7.3. Case Study—Mozambique

This nursery was established in 2018 by ESPANOR (Esperança Para Novo Rebento), an NGO in Milange, Mozambique. Nursery goals are to produce quality seedlings to mitigate deforestation in the Zambezia province, to provide opportunities for local community members and school-aged children to learn technical nursery skills, and to serve as a demonstration nursery for new technologies in forest plant production. The initial container investment to grow 5000 seedlings was $1300 USD for Deepot™ containers (various sizes) and trays plus $2300 USD shipping. After two growing seasons, the D27 size has been designated the preferred size for the species and growing conditions in northern Mozambique. The goal with container selection was to provide the nursery with materials that will last a minimum of 10 years. Additionally, annual substrate cost for the first two years was $570 USD. The nursery space is 160 m$^2$ which is large enough to eventually grow 20,000 or more seedlings per growing season. The growing medium is a mix of locally sourced pine bark (partially composted), coco peat, perlite, and a slow-release fertilizer (SRF). This mix has proven to be very successful in terms of drainage, bulk density, and nutrient levels. SRF is added to the seedlings approximately four months after sowing due to high temperatures and humidity at the nursery site. The presence of weed seed, pathogens, and insects has been very low.

First and second season survival in the nursery was 75% and 70%, respectively. Due to the use of Deepots™, seedling root form has been satisfactory. The root–shoot ratio was lower than desired due to tall seedlings, but irrigation adjustments are being made to correct this. Overall, seedlings have been relatively uniform and healthy.

### 7.4. Case Study—Togo

The nursery was established in 2016 in Notsé, Togo and is a collaboration between the Institute for Community Partnerships and Sustainable Development (a non-profit based in Moscow, ID) and the University of Idaho's Center for Forest Nursery and Seedling Research. The Togo Nursery currently has a capacity of 5000 seedlings and is being used as a demonstration nursery and education site in the greater Togo region. Lolonu, a local women's cooperative group, is being trained to staff and manage the nursery on a day-to-day basis, and local high school and university students have also been involved in nursery construction, production, and outplanting designs meant to test seedling competition and survival rates in restoration conditions. In addition, the nursery is intended to produce

seedlings for agroforestry and reforestation efforts across Togo, with a particular focus on native species. The initial investment for containers and trays was $1270 USD plus $1850 USD shipping. A variety of Deepot™ sizes were ordered with the goal of getting 10 or more years of use out of the containers and trays. The first-year substrate investment was $110 USD. The nursery space is approximately 55 m², just enough for the initial crop of 5000 seedlings.

The setup team made a concerted effort to use local supplies and media components. The initial growing medium was composed of locally produced compost, hand-chipped pea-gravel as a substitute for perlite, locally generated rice hulls, and Osmocote® SRF (provided by the visiting U.S. team). The initial germination, bulk density, and electrical conductivity tests on this medium were satisfactory. However, the rice hulls added too much organic matter to a mix that already included compost, and the size of the pea-gravel pieces was too variable. The compost was produced in a facility dedicated to compost production, and although the individual compost ingredients varied from batch to batch, the production process resulted in a product with consistent texture and nutrient composition. Nonetheless, seedlings tended to have a low root–shoot ratio suggesting excessive nitrogen levels. Furthermore, the locally sourced components introduced various pests, pathogens, or weed seeds that affected the overall seedling quality in the nursery. Thus, adjustments to this medium are being considered. First-year survival in the nursery was approximately 90% although root systems were not as fully formed as desired. The nursery is continuing to refine its culturing practices to improve seedling quality.

## 8. Future Directions

Ample evidence exists regarding the immense economic and ecosystem benefits associated with successful forest restoration. Yet, these benefits (and costs of inaction or failure) are still not reflected in the prices, markets, and policies that shape the choice of nursery containers and associated seedling quality and field performance. Worldwide, the ability to achieve landscape restoration goals depends on sustained nursery support, including access to sufficient and appropriate resources, establishment and maintenance of infrastructure, and education for nursery management and staff [8,59,74–76]. Nurseries with inadequate facilities, supplies, or skilled staff tend to produce poor-quality plants [8,72,75,77,78].

To overcome these issues requires a much more holistic view of the environmental, economic, social, logistic, and cultural elements in the cost–benefit equation that influences decisions and support for nursery production systems. This requires incentives and investments to enable, encourage, and even demand that nursery operators, forest managers, and landholders shift and uphold quality nursery production systems. Sustained and effective support must include communities, agencies, policymakers, and land managers. Social science practices can be used to engage and educate communities about ecosystem services (see, e.g., in [79]) and to help all stakeholders develop a shared social-ecological identity [80]. Encouraging a more holistic relationship with the local ecosystem increases the likelihood that communities will embrace, support, and care for seedling production and field establishment in a more sustainable manner and will develop an appreciation beyond commodities for short-term human uses [81].

Land managers should evaluate nursery production and seedling quality as part of an overall evaluation of outplanting success. Incorporating seedling survival and growth after outplanting into the evaluation avoids the risk that nurseries meet production goals based solely on producing large quantities seedlings without regard to quality. Evaluation of outplanting success can then lead to improvements in the nursery system (including selection of containers and growing media) and seedling quality in subsequent seasons. This cycle of nursery production, outplanting, evaluation, and ensuing improvements is the heart of the target plant concept [12] and is pivotal to the success of global land restoration objectives.

**Author Contributions:** Conceptualization, D.L.H., K.B., L.E. and J.B.F.; writing—original draft preparation, D.L.H., K.B., L.E. and B.L.; writing—review and editing, D.L.H., K.B., L.E., J.B.F., B.L, A.A., A.S.D. All authors have read and agreed to the published version of the manuscript.

**Funding:** This research received no external funding.

**Conflicts of Interest:** The authors declare no conflict of interest.

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
