# Peer review of "The High Cost of the Low-Cost Polybag System: A Review of Nursery Seedling Production Systems"

_land, doi:10.3390/land10080826_

Round 1

Reviewer 1 Report

Overall a good paper. The narrative needs refining to describe the big picture first rather than jumping back and forward through the narrative. The larger issue of global plastics and polybags (throughout many industries) could be discussed. It is important to identify the flow of the narrative and the case studies that can help to illustrate the key points in the article. 

At a more detailed level: 

Lines 53 - 65 have a lot of unsubstantiated claims. In some cases it is difficult to ascertain who are direct planters and indirect planters. In some cases costs can be kept low by direct investment in the planting company that has functional nurseries and can create economies of scale. The costs of tree planting can be confusing as there are many 'tree brokers' who indirectly plant trees through profiting from organizations in the field without adequate technology or communications to attract sponsorship and financial support. This makes it difficult to ascertain the 'true' cost of tree planting since the industry has a significant amount of 'middle men' that are not accounted for. It therefore does not necessarily translate that higher cost = better quality, it is important to look at the seedling supply chain. 

The polybag discussion is an important, but the narrative is a little confusing regarding the the criticism of tree price, then a deep dive into sustainability implications of polybags and alternatives. More attention could be paid to the reasons why plastics are so damaging but also so widespread across many different sectors. The environmental harm of polybags is touched on for the constriction of root systems, but the long term damage to the environment would also be useful. 

The narrative needs refining. From line 256 the reader is brought back to discussions of how economic costs of restoration are high. It would be better to start with the larger narrative - restoration, dependence on plastic and then go down to the smaller details of alternative planting options. So starting with the bigger picture and implications and then look at the alternatives. 

The case study of Lebanon is interesting. The cost structure of the different systems is very helpful to understand how an alternative model could be possible. 

In the conclusion, are there any recommendations for investments? Are there any USDA educational opportunities? How can these techniques be scaled? What would it take to enable wide-scale adoption? Is price, knowledge, availability of alternatives the major barrier? What would it take to get widespread change? 

Reviewer 2 Report

I carefully read the submitted manuscript.  This interesting and worthy manuscript well summarized advantages and disadvantages of modern container system.

I think topic is suitable for Land and this review provide important and useful information for future greening and conservation of vegetation.

Here I suggest some minor comments.

#1, on line 39, I think "[1,2)" should be "[1,2]".

#2, line 225 and others, on the first appearance of each name of modern container system with TM or R mark, I think additional information is helpful for readers. For example, Hiko TM (company name, city, country).

#3 Table3, in Anjar, "Fraxinus angrustifolia" mentioned two times. Also, the Latin names should be listed in alphabetical order. "Acer sp" should be listed as same position in other region.

Finally, I re-confirmed the merits of modern container system of this review.

That's all.

Reviewer 3 Report

The manuscript "The High Cost of the Low-Cost Polybag System: A Review of Nursery Seedling Production Systems" was submitted to Land MDPI. It is a detailed review of the literature on the subject specified in the title of the work, prepared by an international team of authors. The manuscript was supplemented with photographic documentation that corresponds well with the detailed themes discussed. An additional advantage of the study are the tables it contains, which are helpful in tracking the text, and thus have been well prepared. Now, on the text itself: Abstract and Keywords are appropriate. The Introduction is also not objectionable.

  1. On the other hand, in the remaining sections, (i.e. The polybag system, The modern Container System, Comparison Challenges, Economic and Environmental Consequences and Opportunities Associated with Nursery Growing System, Case study 1, Case Study 2, Case Study 3), which were probably written by specific authors without explicit coordination on the part of the corresponding author, so some basic information is repeated. Please, remove this shortcoming.

  1. Conclusions should be placed in points referring to individual parts of the text (freely separated by the authors)
  2. References section: Please determine the entry of journals and check whether, in return for some less important publications, it is not worth indicating works from the last period, let's say seven years.

Overall, I really appreciate the fact that the authors took up such an important topic. The more it is worth refining the manuscript in every detail.

Round 2

Reviewer 1 Report

I think this version is fine for publication